# Numerically Robust Fixed-Point Smoothing Without State Augmentation

**Nicholas Krämer**                                                                     *pekra@dtu.dk*
*Technical University of Denmark*
*Kongens Lyngby, Denmark*

**Reviewed on OpenReview:** *https: // openreview. net/ forum? id= LVQ8BEL5n3*

## Abstract

Practical implementations of Gaussian smoothing algorithms have received a great deal of attention in the last 60 years. However, almost all work focuses on estimating complete time series ("fixed-*interval* smoothing", $\mathcal{O}(K)$ memory) through variations of the Rauch–Tung–Striebel smoother, rarely on estimating the initial states ("fixed-*point* smoothing", $\mathcal{O}(1)$ memory). Since fixed-point smoothing is a crucial component of algorithms for dynamical systems with unknown initial conditions, we close this gap by introducing a new formulation of a Gaussian fixed-point smoother. In contrast to prior approaches, our perspective admits a numerically robust Cholesky-based form (without downdates) and avoids state augmentation, which would needlessly inflate the state-space model and reduce the numerical practicality of any fixed-point smoother code. The experiments demonstrate how a JAX implementation of our algorithm matches the runtime of the fastest methods and the robustness of the most robust techniques while existing implementations must always sacrifice one for the other.

Code: *https://github.com/pnkraemer/code-numerically-robust-fixedpoint-smoother*

## 1 Introduction

Linear Gaussian state-space models and Bayesian filtering and smoothing enjoy numerous applications in areas like tracking, navigation, or control (Grewal and Andrews, 2014; Särkkä and Svensson, 2023). They also serve as the computational backbone of contemporary machine learning methods that revolve around time-series, streaming data, or sequence modelling; for example, message passing, Gaussian processes, and probabilistic numerics (Grewal and Andrews, 2014; Särkkä and Solin, 2019; Murphy, 2023; Hennig et al., 2022). Filtering and smoothing have also been used for constructing and training neural networks (Singhal and Wu, 1988; Gu and Dao, 2023; Chang et al., 2023) or continual learning (Sliwa et al., 2024). All of these applications require fast and robust algorithms for state-space models; we propose a new one in this paper.

Recent developments in Bayesian smoothing focus on subjects like linearisation (e.g. García-Fernández et al., 2016), temporal parallelisation (e.g. Särkkä and García-Fernández, 2020), or numerically robust implementations (e.g. Yaghoobi et al., 2022). However, they all exclusively focus on fixed-*interval* smoothing, which targets the full time-series $p(x_{0:K} \mid y_{1:K})$ in $\mathcal{O}(K)$ memory, never on fixed-*point* smoothing, which only yields the initial value $p(x_0 \mid y_{1:K})$ but does so in $\mathcal{O}(1)$ memory. Even though fixed-point smoothing is a crucial component of, for example, estimating past locations of a spacecraft (Meditch, 1969), it has yet to receive much attention in the literature on state estimation in dynamical systems. The experiments in Section 4 demonstrate fixed-point smoothing applications in probabilistic numerics and in a tracking problem. We anticipate that probabilistic numerical solvers for differential equations will especially benefit from improving the practicality of fixed-point smoothers. The reason is that these algorithms closely connect to filtering and smoothing algorithms (Schober et al., 2019), especially the numerically robust kind (Krämer and Hennig, 2024), and that unknown initial conditions are typical for parameter estimation problems in differential equations and scientific machine learning (Rackauckas et al., 2020). Section 4.2 revisits probabilistic numerics as a prime application for fixed-point smoothing.

**Contributions** We introduce an implementation of numerically robust fixed-point smoothing. Our approach avoids the typical construction of fixed-point smoothers via state-augmentation (Biswas and Mahalanabis, 1972; Smith and Roberts, 1982), which increases the dimension of the state-space model and thus makes estimation needlessly expensive. And unlike previous work on fixed-point smoothing (Meditch, 1967a; Särkkä and Hartikainen, 2010; Rauch, 1963; Meditch, 1967b; 1976; Meditch and Hostetter, 1973; Nishimura, 1969), our perspective is not tied to any particular parametrisation of Gaussian variables. Instead, our proposed algorithm enjoys numerical robustness and compatibility with data streams, while maintaining minimal complexity. There exist tools for each of those desiderata, but our algorithm is the first to deliver them all at once. We demonstrate the algorithm's efficiency and robustness on a sequence of test problems, including a probabilistic numerical method for boundary value problems (Krämer and Hennig, 2021), and show how to use fixed-point smoothing for parameter estimation in Gaussian state-space models.

**Notation** Enumerated sets are abbreviated with subscripts, for example $x_{0:K} \coloneqq \{x_0, ..., x_K\}$ and $y_{1:K} \coloneqq \{y_1, ..., y_K\}$. For sequential conditioning of Gaussian variables (like in the Kalman filter equations), we indicate the most recently visited data points with subscripts in the parameter vectors, for example, $p(x_k \mid y_{1:k-1}) = \mathcal{N}(m_{k|k-1}, C_{k|k-1})$ or $\mathcal{N}(m_{K|K}, C_{K|K}) = p(x_K \mid y_{1:K})$. $I_n$ is the identity matrix with $n$ rows and columns. All covariance matrices shall be symmetric and positive semidefinite. Like existing work on numerically robust state-space model algorithms (e.g. Grewal and Andrews, 2014; Krämer, 2024), we define the *generalised Cholesky factor* $L_\Sigma$ of a covariance matrix $\Sigma$ as any matrix that satisfies $\Sigma = L_\Sigma (L_\Sigma)^\top$. This definition includes the "true" Cholesky factor if $\Sigma$ is positive definite but applies to semidefinite $\Sigma$; for instance, the zero-matrix is the generalised Cholesky factor of the zero-matrix (which does not admit a Cholesky decomposition). There are numerous ways of parametrising multivariate Gaussian distributions, but we exclusively focus on two kinds. Like Yaghoobi et al. (2022), we distinguish:

- *Covariance-based parametrisations:* Parametrise multivariate Gaussian distributions with means and covariance matrices. Covariance-based parametrisations are the standard approach.

- *Cholesky-based parametrisations:* Parametrise multivariate Gaussian distributions with means and generalised Cholesky factors of covariance matrices instead of covariance matrices. Manipulating Gaussian variables in Cholesky-based parametrisations replaces the addition and subtraction of covariance matrices with QR decompositions, which improves the numerical robustness at the cost of a slightly increased runtime; it leads to methods like the square-root Kalman filter (Bennett, 1965).

Other forms, such as the information or canonical form of a multivariate Gaussian distribution (Murphy, 2022), are not directly relevant to this work.

## 2 Problem statement: Fixed-point smoothing

### 2.1 Background on filtering and smoothing

**Linear Gaussian state-space models** This work only discusses linear, discrete state-space models with additive Gaussian noise because such models are the starting point for Bayesian filtering and smoothing. Nonlinear extensions are future work. For integers $d$ and $D$, let $A_1, ..., A_K \in \mathbb{R}^{D \times D}$ and $H_1, ..., H_K \in \mathbb{R}^{d \times D}$ be linear operators and $C_{0|0}, B_1, ..., B_K \in \mathbb{R}^{D \times D}$ and $R_1, ..., R_K \in \mathbb{R}^{d \times d}$ be covariance matrices. Introduce a vector $m_{0|0} \in \mathbb{R}^D$. Assume that observations $y_{1:K}$ are available according to the state-space model (and potentially as a data stream)

$$x_0 = \theta, \quad x_k = A_k x_{k-1} + b_k, \quad y_k = H_k x_k + r_k, \quad k = 1, ..., K, \tag{1}$$

with pairwise independent Gaussian random variables

$$\theta \sim \mathcal{N}(m_{0|0}, C_{0|0}), \quad b_k \sim \mathcal{N}(0, B_k), \quad r_k \sim \mathcal{N}(0, R_k), \quad k = 1, ..., K. \tag{2}$$

In order to simplify the notation, Equations 1 and 2 assume that the Gaussian variables $b_{1:K}$ and $r_{1:K}$ have a zero mean and that there is no observation of the initial state $x_0 = \theta$. The experiments in Section 4

Table 1: *Estimation in Gaussian state-space models.* Other estimation tasks exist, for example, fixed-lag smoothing. However, this article focuses on fixed-point smoothing and its relationship with filtering and fixed-interval smoothing. "RTS smoother": "Rauch–Tung–Striebel smoother".

| Task | Target |
|------|--------|
| Filtering (e.g. Kalman filter) | Real-time updates $\{p(x_k \mid y_{1:k})\}_{k=1}^{K}$ |
| Fixed-interval smoothing (e.g. RTS smoother) | Full time series $p(x_{0:K} \mid y_{1:K})$ |
| Fixed-point smoothing | Initial state $p(x_0 \mid y_{1:K})$ |

demonstrate successful fixed-point smoothing even when violating those two assumptions. Estimating $x_{1:K}$ from observations $y_{1:K}$ is a standard setup for filtering and smoothing (Särkkä and Svensson, 2023).

**Kalman filtering**    Different estimators target different conditional distributions (Table 1). For example, the *Kalman filter* (Kalman, 1960) computes $\{p(x_k \mid y_{1:k})\}_{k=1}^{K}$ by initialising $p(x_0 \mid y_{1:0}) = p(x_0)$ and alternating

$$\text{Prediction:} \qquad p(x_{k-1} \mid y_{1:k-1}) \longmapsto p(x_k \mid y_{1:k-1}) \qquad k = 1, ..., K \qquad (3a)$$

$$\text{Update:} \qquad p(x_k \mid y_{1:k-1}) \longmapsto p(x_k \mid y_{1:k}) \qquad k = 1, ..., K. \qquad (3b)$$

Since all operations are linear and all distributions Gaussian, the recursions are available in closed form. Detailed iterations are in Appendix A. An essential extension of the Kalman filter is the *square-root Kalman filter* (Bennett, 1965; Andrews, 1968) (see also Grewal and Andrews (2014)), which yields the identical distributions as the Kalman filter but manipulates Cholesky factors of covariance matrices instead of covariance matrices. The advantage of the Cholesky-based implementation of the Kalman filter over the covariance-based version is that covariance matrices are guaranteed to remain symmetric and positive semidefinite, whereas accumulated round-off errors can make the covariance-based Kalman filter break down. In this work, we introduce Cholesky-based implementations for fixed-point smoothing, among other things. Both the Kalman filter and its Cholesky-based extension cost $\mathcal{O}(KD^3)$ runtime and $\mathcal{O}(D^2)$ memory, assuming $D \geq d$ (otherwise, exchange $D$ for $d$). The Cholesky-based filter is slightly more expensive because it uses QR decompositions instead of matrix multiplication; Appendix A contrasts Cholesky-based and covariance-based Kalman filtering.

**Fixed-interval smoothing**    While the filter computes the terminal state $p(x_K \mid y_{1:K})$, the *fixed-interval smoother* targets the entire trajectory $p(x_{0:K} \mid y_{1:K})$. For linear Gaussian state-space models, fixed-interval smoothing is implemented by the (fixed-interval) Rauch–Tung–Striebel smoother (Rauch et al., 1965): Relying on independent noise in Equation 2, the Rauch–Tung–Striebel smoother factorises the conditional distribution backwards in time according to (interpret $y_{1:0} = \emptyset$ for notational brevity),

$$p(x_{0:K} \mid y_{1:K}) = p(x_K \mid y_{1:K}) \prod_{k=1}^{K} p(x_{k-1} \mid x_k, y_{1:k-1}). \qquad (4)$$

The first term, $p(x_K \mid y_{1:K})$, is the filtering distribution at the final state and usually computed with a Kalman filter. The remaining terms, $p(x_{k-1} \mid x_k, y_{1:k-1})$, can be assembled with the prediction step in the filtering pass (Equation 3a), which preserves the Kalman filter's $\mathcal{O}(KD^3)$ runtime complexity but increases the memory consumption from $\mathcal{O}(D^2)$ to $\mathcal{O}(KD^2)$. Like the Kalman filter, the Rauch–Tung–Striebel smoother allows covariance- and Cholesky-based parametrisations (Park and Kailath, 1995) in roughly the same complexity. In practice, Cholesky-based arithmetic costs slightly more than covariance-based arithmetic but enjoys an increase in numerical robustness. Appendix B contrasts both implementations.

**Towards fixed-point smoothing**    Finally, we turn to the *fixed-point smoothing* problem: computing the conditional distribution of the initial state $p(x_0 \mid y_{1:K})$ conditioned on all observations (Figure 1). The central difficulty for fixed-point smoothing is that we ask for $\mathcal{O}(K)$ algorithms even though all observations $y_{1:K}$ are in the "future" of $x_0$, which lacks an immediate sequential formulation. That said, one algorithm for sequentially solving the fixed-point smoothing problem involves Rauch–Tung–Striebel smoothing: The fixed-point smoothing problem could be solved by assembling $p(x_{0:K} \mid y_{1:K})$ with the Rauch–Tung–Striebel smoother, which yields a parametrisation of $p(x_0 \mid y_{1:K})$ in closed form. Unfortunately, this solution is not

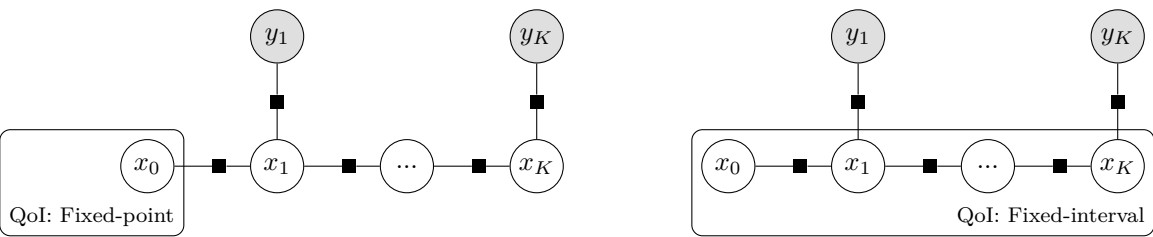

Figure 1: Fixed-point (left) versus fixed-interval smoothing problem (right) as factor graphs. The shaded variables are observed. "QoI": "Quantity of interest".

very efficient: the Rauch–Tung–Striebel smoother stores parametrisations of all conditional distributions $\{p(x_{k-1} \mid x_k, y_{1:k-1})\}_{k=1}^K$, which is problematic for long time-series because it consumes $\mathcal{O}(KD^2)$ memory. In contrast, state-of-the-art fixed-point smoothers require only $\mathcal{O}(D^2)$ memory.

## 2.2 Existing approaches to fixed-point smoothing

The state of the art in fixed-point smoothing does not use fixed-interval smoothers to compute $p(x_0 \mid x_{1:K})$. There are two more efficient approaches: Either we solve the fixed-point smoothing problem by computing the solution to a specific high-dimensional filtering problem (Biswas and Mahalanabis, 1972, "fixed-point smoothing via state-augmented filtering" below), or we derive recursions for how $p(x_0 \mid y_{1:k})$ evolves with $k = 1, ..., K$ (Meditch, 1967b;a; 1969, "fixed-point recursions" below). Both solutions have advantages and disadvantages. We discuss both before explaining how our algorithm fixes their shortcomings.

**Fixed-point smoothing via state-augmented filtering**     Fixed-point smoothing can be implemented with a Kalman filter, more precisely, by running a filter on the state-space model (Biswas and Mahalanabis, 1972)

$$\begin{pmatrix} x_k \\ x_0 \end{pmatrix} = \begin{pmatrix} A_k & 0 \\ 0 & I_D \end{pmatrix} \begin{pmatrix} x_{k-1} \\ x_0 \end{pmatrix} + \begin{pmatrix} I_D \\ 0 \end{pmatrix} b_k, \quad y_k = \begin{pmatrix} H_k & 0 \end{pmatrix} \begin{pmatrix} x_k \\ x_0 \end{pmatrix} + r_k, \quad k = 1, ..., K \tag{5}$$

with initial condition

$$\begin{pmatrix} x_0 \\ x_0 \end{pmatrix} \sim N \left( \begin{pmatrix} m_{0|0} \\ m_{0|0} \end{pmatrix}, \begin{pmatrix} C_{0|0} & C_{0|0} \\ C_{0|0} & C_{0|0} \end{pmatrix} \right) = N \left( \begin{pmatrix} m_{0|0} \\ m_{0|0} \end{pmatrix}, \begin{pmatrix} L_{C_{0|0}} & 0 \\ L_{C_{0|0}} & 0 \end{pmatrix} \begin{pmatrix} L_{C_{0|0}} & 0 \\ L_{C_{0|0}} & 0 \end{pmatrix}^\top \right) \tag{6}$$

The covariance of the initial condition is rank-deficient. However, the generalised Cholesky factor remains well-defined according to Equation 6. The difference between the state-space models in Equation 5 and Equation 1 is that Equation 5 tracks the augmented state $(x_k, x_0)$ instead of $x_k$. As a result, the Kalman filter applied to Equations 5 and 6 yields parametrisations of the conditional distributions $\{p(x_k, x_0 \mid y_{1:k})\}_{k=1}^K$ in $\mathcal{O}(K(2D)^3)$ runtime and $\mathcal{O}((2D)^2)$ memory. The factor "$2D$" stems from doubling the state-space dimension. The initial condition $p(x_0 \mid y_{1:K})$ emerges from the augmented state $p(x_K, x_0 \mid y_{1:K})$ in closed form. Relying on the Kalman filter formulation inherits all computational benefits of Kalman filters, most importantly, the Kalman filter's Cholesky-based form with its attractive numerical robustness. Nonetheless, doubling of the state-space dimension is unfortunate because it increases the runtime roughly by factor 8 (because $(2D)^3 = 8D^3$) and the memory by factor 4 (because $(2D)^2 = 4D^2$). However, as long as one commits to covariance-based arithmetic, this increase can be avoided:

**Fixed-point recursions**     Suppose we temporarily ignore Cholesky-based forms and commit to covariance-based parametrisations of Gaussian distributions. In that case, we can improve the efficiency of the fixed-point smoother. It turns out that during the forward pass, the mean and covariance of $p(x_0 \mid y_{1:k})$ follow a recursion that can be run parallel to the Kalman filter pass (Meditch, 1969; Särkkä and Hartikainen, 2010; Meditch, 1967b). Särkkä and Hartikainen (2010) explain how to implement this recursion: start by initialising $m_{0|0}$ and $C_{0|0}$ according to Equation 1 and set $G_{0|0} = I$. Then, iterate from $(m_{0|k-1}, C_{0|k-1}, G_{0|k-1})$ to

Table 2: Existing approaches to the fixed-point smoothing problem. The "fixed-point recursion" represents Meditch (1969; 1967b) through Equation 7. "$\mathcal{O}(1)$ memory" stands in contrast to the $\mathcal{O}(K)$ memory of a fixed-interval smoother, and the "low-dimensional state" refers to avoiding state-augmentation, which affects both runtime and memory because it doubles the dimension of the state-space model.

| Method | $\mathcal{O}(1)$ memory | Low-dimensional state | Cholesky-based |
|---|---|---|---|
| Via Rauch–Tung–Striebel smoothing | ✗ | ✓ | ✓ |
| Via filtering with augmented state | ✓ | ✗ | ✓ |
| Fixed-point recursion (Equation 7) | ✓ | ✓ | ✗ |
| This work (Algorithms 1 to 3) | ✓ | ✓ | ✓ |

$(m_{0|k}, C_{0|k}, G_{0|k})$ using the recursion (Särkkä and Hartikainen, 2010; Särkkä, 2013)

$$G_{0|k} = G_{0|k-1}G_{k-1|k} \tag{7a}$$

$$m_{0|k} = m_{0|k-1} + G_{0|k}(m_{k|k} - m_{k|k-1}) \tag{7b}$$

$$C_{0|k} = C_{0|k-1} + G_{0|k}(C_{k|k} - C_{k|k-1})(G_{0|k})^\top. \tag{7c}$$

Next to $(m_{0|k-1}, C_{0|k-1}, G_{0|k-1})$, these formulas only depend on the output of the prediction step as well as the smoothing gains $G_{k-1|k}$ (smoothing gains: Appendix B). For $k = 1$, the fixed-point gain intentionally equals the smoothing gain, which Equation 7 expresses by initialising $G_{0|0} = I$. The recursion in Equation 7 can be implemented to run simultaneously with the forward filtering pass. Unfortunately, even though this implementation avoids doubling the size of the state-space model and enjoys $\mathcal{O}(D^2)$ memory and $\mathcal{O}(KD^3)$ runtime, a Cholesky-based formulation has been unknown (until now). Thus, Equation 7 cannot be applied to problems where numerical robustness is critical, like the probabilistic numerical simulation of differential equations (Krämer, 2024). The main contribution of this paper is to generalise Equation 7 to enable Cholesky-based parametrisations:

**Contribution.** *Compute the solution of the fixed-point smoothing problem $p(x_0 \mid y_{1:K})$ in $\mathcal{O}(KD^3)$ runtime, $\mathcal{O}(D^2)$ memory, using any parametrisation of Gaussian variables, and without state augmentation; Table 2.*

## 3 The method: Numerically robust fixed-point smoothing

Our approach to numerically robust fixed-point smoothing involves two steps: First, we derive a recursion for the conditional distribution $p(x_0 \mid x_k, y_{1:k})$ instead of one for the joint distribution $p(x_0, x_k \mid y_{1:k})$, $k = 1, ..., K$. Second, we implement the recursion in Cholesky-based parametrisations without losing closed-form, constant-memory updates. As a byproduct of this derivation, other parametrisations of Gaussian distributions (like the information form) become possible, too. However, we focus on Cholesky-based implementations for their numerical robustness and leave other choices to future work.

### 3.1 A new recursion for fixed-point smoothing

A derivation of a new fixed-point smoothing recursion follows. Similar to the state-augmented filtering perspective, the target distribution $p(x_0 \mid y_{1:K})$ can be written as the marginal of the augmented state

$$p(x_0 \mid y_{1:K}) = \int p(x_0, x_K \mid y_{1:K}) \, \mathrm{d}x_K. \tag{8}$$

Computing $p(x_0 \mid y_{1:K})$ becomes a byproduct of computing $p(x_0, x_K \mid y_{1:K})$, and the latter admits a sequential formulation. This perspective was essential to implementing fixed-point smoothing with state-augmented filtering. The augmented state factorises into the conditional

$$p(x_0, x_K \mid y_{1:K}) = p(x_0 \mid x_K, y_{1:K})p(x_K \mid y_{1:K}). \tag{9}$$

Since $p(x_K \mid y_{1:K})$ is already computed in the forward pass, it suffices to derive a recursion for the fixed-point conditional $p(x_0 \mid x_K, y_{1:K})$. The backward factorisation of the smoothing distribution in Equation 4 implies

$$p(x_0 \mid x_K, y_{1:K}) = \int \prod_{k=1}^{K} p(x_{k-1} \mid x_k, y_{1:k-1}) \, dx_{1:K-1}. \tag{10}$$

Marginalising over all intermediate states $x_{1:K-1}$ like in Equation 10 turns a Rauch–Tung–Striebel smoother into a fixed-point smoother. An $\mathcal{O}(K)$ runtime implementation now emerges from observing how the integral in Equation 10 rearranges to a nested sequence of single-variable integrals,

$$p(x_0 \mid x_K, y_{1:K}) = \int \prod_{k=1}^{K} p(x_{k-1} \mid x_k, y_{1:k-1}) \, dx_{1:K-1} \tag{11a}$$

$$= \int \dots \left( \int \left[ \int p(x_0 \mid x_1) p(x_1 \mid x_2, y_{1:1}) \, dx_1 \right] p(x_2 \mid x_3, y_{1:2}) \, dx_2 \right) \dots dx_{K-1} \tag{11b}$$

$$= \int \dots \left( \int p(x_0 \mid x_2, y_1) p(x_2 \mid x_3, y_{1:2}) \, dx_2 \right) \dots dx_{K-1} \tag{11c}$$

This rearranging of the integrals is essential to the derivation because it implies a forward-in-time recursion:

**Algorithm 1** (Fixed-point smoother). *To compute the solution to the fixed-point smoothing problem, assemble $p(x_K \mid y_{1:K})$ with a Kalman filter and evaluate $p(x_0 \mid x_K, y_{1:K})$ as follows. (To simplify the index-related notation in this algorithm, read $y_{1:-1} = y_{1:0} = \emptyset$.)*

1. *Initialise $p(x_0 \mid x_0, y_{1:-1}) = \mathcal{N}(G_{0|0}x_0 + p_{0|0}, P_{0|0})$ with $G_{0|0} = I_D$, $p_{0|0} = 0$, and $P_{0|0} = 0$.*

2. *For $k = 1, \dots, K$, iterate from $k-1$ to $k$,*

$$p(x_0 \mid x_k, y_{1:k-1}) = \int p(x_0 \mid x_{k-1}, y_{1:k-2}) p(x_{k-1} \mid x_k, y_{1:k-1}) \, dx_{k-1}. \tag{12}$$

*The conditionals $p(x_{k-1} \mid x_k, y_{1:k-1})$ are from the Rauch–Tung–Striebel smoother (Equation 4).*

3. *Marginalise $p(x_0 \mid y_{1:K})$ from $p(x_K \mid y_{1:K})$ and $p(x_0 \mid x_K, y_{1:K-1})$ via Equation 8.*

Equation 12 turns a Rauch–Tung–Striebel smoother into a fixed-point smoother: The recursion requires access to $p(x_{k-1} \mid x_k, y_{1:k-1})$ computed by the forward-pass of the Rauch–Tung–Striebel smoother. Therefore, Equation 12 runs concurrently to the forward filtering pass.

## 3.2 Cholesky-based implementation

The missing link for Algorithm 1 is the merging of two affinely related conditionals in Equation 12. Since both conditionals in Equation 12 are affine and Gaussian, their combination $p(x_0 \mid x_k, y_{1:k-1})$ is Gaussian and available in closed form for all $k = 1, \dots, K$. Its parametrisation depends on the parametrisation of each input distribution. For covariance- and Cholesky-based arithmetic, it looks as follows:

**Algorithm 2** (Covariance-based implementation of Equation 12). *Recall the initialisation of Algorithm 1 and the convention $y_{1:-1} = y_{1:0} = \emptyset$. For any $k = 1, \dots, K$, if the fixed-point and smoothing conditionals (see Equation 4 & Appendix B for the latter) are parametrised by*

$$p(x_0 \mid x_{k-1}, y_{1:k-2}) = \mathcal{N}(G_{0|k-1}x_{k-1} + p_{0|k-1}, P_{0|k-1}) \tag{13a}$$

$$p(x_{k-1} \mid x_k, y_{1:k-1}) = \mathcal{N}(G_{k-1|k}x_k + p_{k-1|k}, P_{k-1|k}), \tag{13b}$$

*for given $G_{i|j} \in \mathbb{R}^{D \times D}$, $p_{i|j} \in \mathbb{R}^D$, and $P_{i|j} \in \mathbb{R}^{D \times D}$, then, the next fixed-point conditional is*

$$p(x_0 \mid x_k, y_{1:k-1}) = \mathcal{N}(G_{0|k}x_k + p_{0|k}, P_{0|k}), \tag{14a}$$

$$G_{0|k} := G_{0|k-1}G_{k-1|k}, \tag{14b}$$

$$p_{0|k} := G_{0|k-1}p_{k-1|k} + p_{0|k-1}, \tag{14c}$$

$$P_{0|k} := G_{0|k-1}P_{k-1|k}(G_{0|k-1})^\top + P_{0|k-1}. \tag{14d}$$

*Implementing this operation costs $\mathcal{O}(D^3)$ floating-point operations.*

**Algorithm 3** (Cholesky-based implementation of Equation 12). *For any $k = 1, ..., K$ (again $y_{1:-1} = y_{1:0} = \emptyset$), if the previous fixed-point conditional and smoothing transition (see Equation 4 & Appendix B) are*

$$p(x_0 \mid x_{k-1}, y_{1:k-2}) = \mathcal{N}(G_{0|k-1}x_{k-1} + p_{0|k-1}, L_{P_{0|k-1}}(L_{P_{0|k-1}})^\top) \tag{15a}$$

$$p(x_{k-1} \mid x_k, y_{1:k-1}) = \mathcal{N}(G_{k-1|k}x_k + p_{k-1|k}, L_{P_{k-1|k}}(L_{P_{k-1|k}})^\top), \tag{15b}$$

*for given $G_{i|j} \in \mathbb{R}^{D \times D}$, $p_{i|j} \in \mathbb{R}^D$, and $L_{P_{i|j}} \in \mathbb{R}^{D \times D}$, then, the next fixed-point conditional is*

$$p(x_0 \mid x_k, y_{1:k-1}) = \mathcal{N}(G_{0|k}x_k + p_{0|k}, L_{P_{0|k}}(L_{P_{0|k}})^\top), \tag{16a}$$

$$G_{0|k} := G_{0|k-1}G_{k-1|k}, \tag{16b}$$

$$p_{0|k} := G_{0|k-1}p_{k-1|k} + p_{0|k-1}, \tag{16c}$$

$$L_{P_{0|k}} := \mathfrak{R}^\top, \tag{16d}$$

*where $\mathfrak{R}$ is the upper triangular matrix returned by the QR-decomposition*

$$\mathfrak{Q}\,\mathfrak{R} = \begin{pmatrix} (L_{P_{k-1|k}})^\top (G_{0|k-1})^\top \\ (L_{P_{0|k-1}})^\top \end{pmatrix}. \tag{17}$$

*Implementing this operation costs $\mathcal{O}(D^3)$ floating-point operations.*

Algorithm 2 follows from the rules for manipulating affinely related Gaussian distributions (e.g. Särkkä, 2013, Appendix A.1). Algorithm 3 is indeed the Cholesky-based version of Algorithm 2: The expression in Equation 16d is the generalised Cholesky factor of the expression in Equation 14d because

$$L_{P_{k-1|k}}(L_{P_{k-1|k}})^\top = \mathfrak{R}^\top \mathfrak{Q}^\top \mathfrak{Q}\mathfrak{R} = \begin{pmatrix} G_{0|k-1}L_{P_{k-1|k}} & L_{P_{0|k-1}} \end{pmatrix} \begin{pmatrix} (L_{P_{k-1|k}})^\top (G_{0|k-1})^\top \\ (L_{P_{0|k-1}})^\top \end{pmatrix} = P_{k-1|k} \tag{18}$$

holds. The combination of Algorithms 1 and 3 is a novel, Cholesky-based implementation of a fixed-point smoother. Similarly, the combination of Algorithms 1 and 2 is a novel, covariance-based implementation of a fixed-point smoother. With a small modification to Algorithm 1, combining Algorithm 2 with the covariance-based formulas in Algorithm 1 recovers Equation 7:

**Proposition 1.** *If the combination of Algorithms 1 and 2 computes the marginals*

$$p(x_0 \mid y_{1:k}) = \int p(x_0 \mid x_k, y_{1:k-1})p(x_k \mid y_{1:k}) \, dx_k, \tag{19}$$

*at every $k = 1, ..., K$ instead of only at the final time-step, the recursion reduces to Equation 7.*

*Proof.* Induction; Appendix C. □

**Computational complexity** Both Algorithms 2 and 3 cost $\mathcal{O}(D^3)$ floating-point operations per step. The $k$th iteration in Algorithm 1 needs access to the same quantities as the $k$th iteration of the forward pass of a Rauch–Tung–Striebel smoother, which can be implemented in $\mathcal{O}(D^2)$ memory. Unlike the Rauch–Tung–Striebel smoother, Algorithm 1 does not store intermediate results, which is why it consumes $\mathcal{O}(D^2)$ memory in total (instead of $\mathcal{O}(KD^2)$). Both Algorithm 2 and Equation 7 require three matrix-matrix- and one matrix-vector multiplication per step, so they are equally efficient (Algorithm 2 uses fewer matrix additions. However, the addition of matrices is fast). Therefore, it will be no loss of significance that we always implement covariance-based fixed-point smoothing via Algorithm 1 instead of Equation 7 in our experiments.

| | Covariance-based | Cholesky-based | |
|---|---|---|---|
| **Via filter** | Known | Known | |
| **Via Rauch–Tung–Striebel** | Known | Known | |
| **Fixed-point recursion** | Known, but see Prop. 1 | Our contribution | Experiment II: Robustness |
| | | Experiment I: Efficiency | |

Figure 2: Outline of the memory-, runtime-, and robustness-related demonstrations.

## 4    Experiments

The experiments serve two purposes. To start with, they investigate whether the proposed Cholesky-based implementation (Algorithm 3) of the fixed-point smoother recursion (Algorithm 1) holds its promises about memory, runtime, and numerical robustness. According to the theory in Section 3, we should observe:

- *Memory:* Slightly lower memory demands than a state-augmented Kalman filter; drastically lower memory demands than a Rauch–Tung–Striebel smoother.

- *Wall-time:* Faster than a state-augmented, Cholesky-based Kalman filter; roughly as fast as a Rauch–Tung–Striebel smoother.

- *Numerical robustness:* Combining Algorithm 1 with Cholesky-based parametrisations (Algorithm 3) is significantly more robust than combining it with covariance-based parametrisations (Algorithm 2); comparable to a Cholesky-based implementation of a Kalman filter.

These three phenomena will be studied in two experiments, one for runtime/memory and one for numerical robustness (outline: Figure 2). The problem set for these two experiments includes a toy problem for the former and an application in probabilistic numerics for the latter. Afterwards, a case study in tracking shows how to use the fixed-point smoother for estimating the initial parameter in a state-space model.

**Hardware and code**    All experiments run on the CPU of a consumer-grade laptop and finish within a few minutes. Our JAX implementation (Bradbury et al., 2018) of Kalman filters, Rauch–Tung–Striebel smoothers, and fixed-point smoothers is at

*https://github.com/pnkraemer/code-numerically-robust-fixedpoint-smoother*

We implement the existing fixed-point smoother recursions in Equation 7 by combining Algorithm 1 with covariance-based parametrisations; recall Proposition 1.

### 4.1    Experiment I: How efficient is the fixed-point recursion?

**Motivation**    Sections 2.2 and 3 mention three approaches to fixed-point smoothing: a detour via Rauch–Tung–Striebel smoothing, state-augmented Kalman filtering, and our recursion in Algorithm 1. In theory, Algorithm 1 should be the most efficient: it does not inflate the state-space model like a state-augmented Kalman filter does and requires $\mathcal{O}(1)$ instead of $\mathcal{O}(K)$ memory, unlike the Rauch–Tung–Striebel smoother. This first experiment demonstrates that these effects are visible in practice.

**Problem setup**    In this first example, we only measure the execution time and memory requirement of three approaches to fixed-point smoothing. Both memory and runtime depend only on the size of a state-space model and not on the difficulty of the estimation task. Therefore, we consider a state-space model where all system matrices and vectors are populated with random values.

**Implementation**    We choose $K = 1,000$, vary $d$, set the size of the hidden state to $D = 2d$, and use Cholesky-based arithmetic for all estimators. We take Equation 1 and introduce a nonzero bias in all noises

Table 3: Runtime in seconds (wall-time, best of three runs). Lower is better. The column-wise lowest are bold & shaded. Randomly populated model. $K = 1,000$ steps. All methods use Cholesky-based parametrisations.

| | $d = 2$ | $d = 5$ | $d = 10$ | $d = 20$ | $d = 50$ | $d = 100$ |
|---|---|---|---|---|---|---|
| Via Rauch–Tung–Striebel | $5.8 \times 10^{-3}$ | $\mathbf{1.8 \times 10^{-2}}$ | $6.5 \times 10^{-2}$ | $4.5 \times 10^{-1}$ | $\mathbf{2.9 \times 10^{0}}$ | $\mathbf{1.2 \times 10^{1}}$ |
| Via filter | $\mathbf{5.0 \times 10^{-3}}$ | $2.1 \times 10^{-2}$ | $8.0 \times 10^{-2}$ | $7.1 \times 10^{-1}$ | $4.6 \times 10^{0}$ | $1.6 \times 10^{1}$ |
| Algorithm 1 | $6.4 \times 10^{-3}$ | $\mathbf{1.8 \times 10^{-2}}$ | $\mathbf{6.4 \times 10^{-2}}$ | $\mathbf{4.4 \times 10^{-1}}$ | $\mathbf{2.9 \times 10^{0}}$ | $\mathbf{1.2 \times 10^{1}}$ |

Table 4: Memory in bytes (we use 32-bit arithmetic). Lower is better. The column-wise lowest entries are bold and shaded. Randomly populated model. $K = 1,000$ steps. All methods use Cholesky-based parametrisations.

| | $d = 2$ | $d = 5$ | $d = 10$ | $d = 20$ | $d = 50$ | $d = 100$ |
|---|---|---|---|---|---|---|
| Via Rauch–Tung–Striebel | $2.2 \times 10^{5}$ | $1.2 \times 10^{6}$ | $4.9 \times 10^{6}$ | $1.9 \times 10^{7}$ | $1.2 \times 10^{8}$ | $4.8 \times 10^{8}$ |
| Via filter | $2.8 \times 10^{2}$ | $1.6 \times 10^{3}$ | $6.5 \times 10^{3}$ | $2.5 \times 10^{4}$ | $1.6 \times 10^{5}$ | $6.4 \times 10^{5}$ |
| Algorithm 1 | $\mathbf{2.2 \times 10^{2}}$ | $\mathbf{1.2 \times 10^{3}}$ | $\mathbf{4.9 \times 10^{3}}$ | $\mathbf{1.9 \times 10^{4}}$ | $\mathbf{1.2 \times 10^{5}}$ | $\mathbf{4.8 \times 10^{5}}$ |

in Equation 2. Then, we randomly populate all system matrices in the state-space model with independent samples from $\mathcal{N}(0, {}^{1}/_{K^2})$. Afterwards, we sample $y_{1:K}$ from this state-space model to generate toy data. A $K$-dependent covariance controls that the samples $y_{1:K}$ remain finite in 32-bit floating-point arithmetic. However, the precise values of the model and data do not matter for this experiment – only their size does. Finally, we vary $d$ and measure the runtime and memory requirements of each of the three methods. To measure runtime, we use wall time in seconds. We display the best of three runs instead of the average because all codes are deterministic (no data-dependence, no randomness), thus the main driver for runtime differences is background machine noise. Selecting the best of three runs minimises this noise as much as possible. To measure memory consumption, we count the number of floating-point values the estimators carry from step to step, multiply this by 32 (because we use 32-bit arithmetic), and translate bits into bytes.

**Evaluation**    The runtime results are in Table 3 and the memory results in Table 4. The runtime data shows that except for $d = 2$, the Rauch–Tung–Striebel smoother and Algorithm 1 are faster than the state-augmented filter. Algorithm 1 is marginally faster than the Rauch–Tung–Striebel smoother code, even though it computes strictly more at every iteration. However, the Rauch–Tung–Striebel smoother executes two loops, one forwards and one backwards, whereas Algorithm 1 only executes a forward loop. This discrepancy might lead to the performance improvement from Rauch–Tung–Striebel smoothing to Algorithm 1. The memory data shows that the Rauch–Tung–Striebel smoothing solution and the Algorithm 1 have identical memory requirements per step. Both consume less storage than the state-augmented filter (per step). As expected, the Rauch–Tung–Striebel smoother requires exactly $K$-times the memory of Algorithm 1, which would be infeasible for long time series over high-dimensional states. In general, this experiment underlines the claimed efficiency of our new fixed-point smoother recursion in a Cholesky-based parametrisation. The following experiment will answer the question of whether Cholesky-based arithmetic is necessary.

## 4.2   Experiment II: How much more robust is the Cholesky-based code?

**Motivation**    Cholesky-based implementations replace matrix addition and matrix multiplication with QR decompositions. They are more expensive than covariance-based implementations because the decomposition is more expensive than matrix multiplication. However, there are numerous situations where the gains in numerical robustness are worth the increase in runtime. One example is the probabilistic numerical simulation of differential equations, which use integrated Wiener processes (Schober et al., 2014; 2019) with high order and small time-steps. For these applications, Cholesky-based implementations are the only feasible approach (Krämer and Hennig, 2024). This second experiment shows that Algorithm 1 unlocks fixed-point smoothing for probabilistic numerical simulation through Algorithm 3.

**Problem setup**     We solve a boundary value problem based on an ordinary differential equation. More specifically, we solve the 15th in the collection of test problems by Mazzia and Cash (2015) (Figure 3),

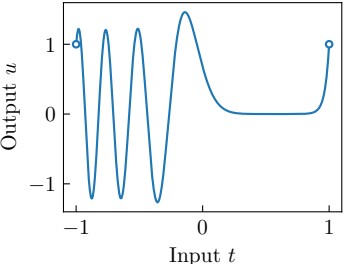

$$10^{-3} \cdot \frac{d^2}{dt^2} u(t) = t u(t), \quad u(-1) = u(1) = 1. \tag{20}$$

We follow the procedure for solving boundary value problems with a probabilistic numerical method by Krämer and Hennig (2021) for the most part. However, we skip the iterated Kalman smoother because the differential equation is linear and skip mesh refinement and expectation maximisation to keep this demonstration simple. The focus lies on numerical robustness. We choose a twice-integrated Wiener process prior and discretise it on $K$ equispaced points in $[-1, 1]$, which yields the latent transitions (let $\Delta t \coloneqq 1/K$)

Figure 3: 15th Boundary value problem (Mazzia and Cash, 2015).

$$A_k \coloneqq \begin{pmatrix} 1 & \Delta t & \frac{(\Delta t)^2}{2} \\ 0 & 1 & \Delta t \\ 0 & 0 & 1 \end{pmatrix}, \quad B_k \coloneqq \begin{pmatrix} \frac{(\Delta t)^5}{20} & \frac{(\Delta t)^4}{8} & \frac{(\Delta t)^3}{3} \\ \frac{(\Delta t)^4}{8} & \frac{(\Delta t)^3}{3} & \frac{(\Delta t)^2}{2} \\ \frac{(\Delta t)^3}{3} & \frac{(\Delta t)^2}{2} & \Delta t \end{pmatrix}, \quad k = 1, ..., K. \tag{21}$$

The dynamics are constant because the grid points are evenly spaced. The means of the process noise are zero, like in Equation 2. The hidden state

$$x_k \coloneqq \left( u(t_k), \frac{d}{dt} u(t_k), \frac{d^2}{dt^2} u(t_k) \right), \quad k = 1, ..., K \tag{22}$$

tracks the differential equation solution $u$ and its derivatives, including $\frac{d^2}{dt^2} u$, which means that the residual $10^{-3} \frac{d^2}{dt^2} u - t u(t)$ is a linear function of $x_k$. We introduce the model for the constraints ($\beta_k$ shall be the nonzero mean of the observation noise $r_k$ in Equations 1 and 2)

$$H_k = \begin{pmatrix} -t_k & 0 & 10^{-3} \end{pmatrix}, \qquad \beta_k = 0, \qquad R_k = 0, \qquad k = 1, ..., K-1, \tag{23a}$$

$$H_K = \begin{pmatrix} 1 & 0 & 0 \end{pmatrix}, \qquad \beta_K = -1, \qquad R_K = 0. \tag{23b}$$

Note how the constraint at the final time-point encodes the right-hand side boundary condition, not the differential equation. We choose the initial mean $m_{0|0} = (1, 0, 0)$ and initial covariance $C_{0|0} = \text{diag}(0, 1, 1)$ to represent the left-hand side boundary condition. Estimating $x_{0:K}$ from $y_{1:K} \coloneqq 0$ solves the boundary value problem ((Krämer and Hennig, 2021); Figure 3 displays the mean of the fixed-interval smoothing solution). Krämer and Hennig (2021) explain how estimating the initial condition $p(x_0 \mid y_{1:K})$ is important for parameter estimation problems. In other words, fixed-point smoothers are relevant for boundary value problem simulation. We are the first to use them for this task.

**Implementation**     For this demonstration, we consider the Cholesky-based implementation of the Kalman filter as the gold standard for numerical robustness because it has been used successfully in numerous similar problems (Krämer and Hennig, 2024; Bosch et al., 2021; Krämer and Hennig, 2021; Krämer, 2024). The Cholesky-based, state-augmented Kalman filter provides a reference solution of the fixed-point equations. We run Cholesky-based (Algorithm 3) and covariance-based code (Algorithm 2) for the fixed-point smoother recursions (Algorithm 1) and measure how much the estimated initial means deviate from the Kalman-filter reference in terms of the absolute root-mean-square error. In infinite precision, the results would be identical. In finite precision, all deviations should be due to a loss of stability. We vary $K$ to investigate how the step-size $\Delta t$ affects the results, with the intuition that smaller steps lead to worse conditioning in $B_k$ and that this effect makes the estimation task more difficult (Krämer and Hennig, 2024).

**Evaluation**     The results are in Table 5. They indicate how the Cholesky-based code is significantly more robust than the covariance-based code. The covariance-based implementation delivers only three meaningful digits for $K = 10$ points, diverges for more grid points, and breaks for $K \geq 500$. In contrast, the Cholesky-based code delivers meaningful approximations across all grids. This consistency underlines how much more robust our Cholesky-based code is and how it unlocks fixed-point smoothing for probabilistic numerics.

Table 5: Deviation from the Cholesky-based, state-augmented Kalman filter on the boundary value problem. Lower is better and close to machine-precision desirable. The column-wise lowest values are bold and coloured. "$K$": number of grid points. Double precision (64-bit arithmetic).

| *-based: | $K = 10$ | $K = 20$ | $K = 50$ | $K = 100$ | $K = 200$ | $K = 500$ | $K = 1,000$ |
|---|---|---|---|---|---|---|---|
| Covariance | $2.9 \times 10^{-3}$ | $3.4 \times 10^{-1}$ | $1.1 \times 10^{0}$ | $1.0 \times 10^{1}$ | $2.1 \times 10^{1}$ | NaN | NaN |
| Cholesky | $\mathbf{2.0 \times 10^{-10}}$ | $\mathbf{5.0 \times 10^{-8}}$ | $\mathbf{4.2 \times 10^{-7}}$ | $\mathbf{7.9 \times 10^{-8}}$ | $\mathbf{1.3 \times 10^{-7}}$ | $\mathbf{6.1 \times 10^{-8}}$ | $\mathbf{3.4 \times 10^{-8}}$ |

### 4.3 Case study: Estimating parameters of a state-space model

**Movitation** The previous two experiments have emphasized the practicality of our proposed method. We conclude by applying fixed-point smoothing to parameter estimation in a tracking model. This study aims to emulate applications of fixed-point smoothing in navigation tasks (Meditch, 1969) in a setup that is easy to reproduce. However, the demonstration also links to the previous boundary value problem experiment through expectation maximisation (Krämer and Hennig, 2021).

**Problem setup** The task is to estimate the mean parameter of an unknown initial condition in a car tracking example. Define the Wiener velocity model on $K = 10$ equispaced points ($\Delta t = 1/10$),

$$A_k := \begin{pmatrix} I_2 & \Delta t \cdot I_2 \\ 0 & I_2 \end{pmatrix} \quad B_k := \begin{pmatrix} \frac{(\Delta t)^3}{3} \cdot I_2 & \frac{(\Delta t)^2}{2} \cdot I_2 \\ \frac{(\Delta t)^2}{2} \cdot I_2 & \Delta t \cdot I_2 \end{pmatrix}, \quad H_k = \begin{pmatrix} I_2 & 0 \end{pmatrix}, \quad R_k = 0.1^2 \cdot I_2. \tag{24}$$

All biases are zero, $b_k = 0$, $r_k = 0$. Textbooks on Bayesian filtering and smoothing (Särkkä, 2013; Särkkä and Svensson, 2023) use this Wiener velocity model to estimate the trajectory of a car. We populate $m$ and $L_C$ with samples from a standard normal distribution and sample an initial condition

$$x_0 = \begin{pmatrix} \theta_1 & \theta_2 & \dot{\theta}_1 & \dot{\theta}_2 \end{pmatrix} \sim \mathcal{N}(m_{0|0}, L_{C_{0|0}}(L_{C_{0|0}})^\top). \tag{25}$$

We sample artificial observations $y_{1:K}$. From here on, the initial mean $m_{0|0}$ will be treated as unknown.

**Implementation** We combine fixed-point smoothing with expectation maximisation (Dempster et al., 1977) to calibrate $m_{0|0}$ using the data $y_{1:K}$. The expectation maximisation update for the initial mean in a linear Gaussian state-space model is $m_{\text{new}} := m_{0|K}$ (Särkkä and Svensson, 2023), and this update repeats until convergence. Here, $m_{0|K}$ is the mean of $p(x_0 \mid y_{1:K})$. We implement the fixed-point smoother recursion in Cholesky-based arithmetic and run expectation maximisation for three iterations. We initialise the mean guess by sampling all entries independently from a centred normal distribution with a variance of 100. We track the data's marginal likelihood ("evidence"), computing it online during the forward filtering pass.

**Evaluation** The results are in Figure 4. They show how the combination of expectation maximisation with fixed-point smoothing recovers the initial mean already after three iterations. The evidence increases at every iteration, but that is normal for expectation maximisation (Wu, 1983). In conclusion, fixed-point smoothing is a viable parameter estimation technique in a state-space model.

## 5 Discussion

**Limitations and future work** Our new approach to fixed-point smoothing strictly generalises existing techniques, but inherits some of their limitations: Even though the new recursion in Algorithm 1 is independent of the type of state-space model, implementing the fixed-point smoother in closed form assumes a linear Gaussian setup. Future work should explore robust fixed-point smoothing in nonlinear state-space models, for example, through posterior linearisation (García-Fernández et al., 2016). Like all Gaussian smoothing algorithms, our methods have cubic complexity in the state-space dimension because of the matrix-matrix arithmetic or QR decompositions, respectively. Finally, other parametrisations of Gaussian variables could be used with Algorithm 1 instead of covariance or Cholesky-based parametrisations (Algorithms 2 and 3); for example, the information form or ensembles (Murphy, 2022; Houtekamer and Mitchell, 2005). Combining these alternatives with numerically robust fixed-point smoothing could be interesting avenues for future work.

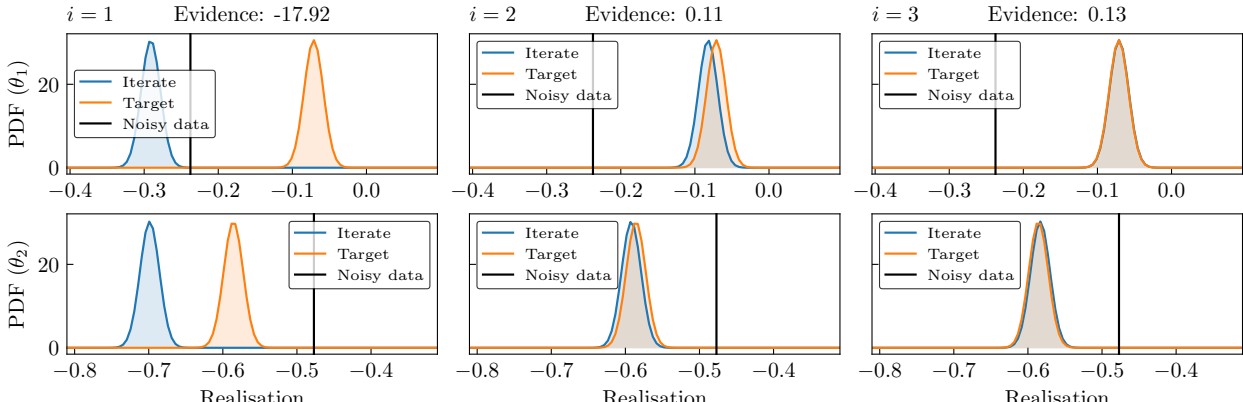

Figure 4: Initial distributions $p(\theta_1, \theta_2 \mid y_{1:K})$ of the car tracking model after running the fixed-point smoother (recall $x := (\theta_1, \theta_2, \dot{\theta}_1, \dot{\theta}_2)$). Left to right: After three iterations, the combination of expectation maximisation with the fixed-point smoother finds the correct initial mean $\theta = (\theta_1, \theta_2)$ of the state-space model. Top: First coordinate $p(\theta_1 \mid y_{1:K})$. Bottom: second coordinate $p(\theta_2 \mid y_{1:K})$. "PDF": "Probability density function".

**Conclusion**     This paper presented a new recursion for fixed-point smoothing in Gaussian state-space models: Algorithm 1. It has lower memory consumption than existing approaches because it avoids state-augmentation and foregoes storing all intermediate results of a Rauch–Tung–Striebel smoother. It also allows arbitrary parametrisations of Gaussian distributions, and we use this perspective to derive a Cholesky-based form of fixed-point smoothers. As a result, our method matches the speed of the fastest and the robustness of the most robust methods, as has been demonstrated in three simulations of varying difficulty. Through these successes, our contribution hopefully revives fixed-point smoothing as part of the literature on Gaussian filters and smoothers. We anticipate notable performance improvements for algorithms at the interface of smoothing and dynamical systems due to the newfound ability to reliably use non-standard smoothing algorithms.

### Acknowledgements

This work was supported by a research grant (42062) from VILLUM FONDEN. The work was partly funded by the Novo Nordisk Foundation through the Center for Basic Machine Learning Research in Life Science (NNF20OC0062606). This project received funding from the European Research Council (ERC) under the European Union's Horizon programme (grant agreement 101125993).

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

## A    Kalman filter recursions

The Kalman filter (Kalman, 1960) computes $p(x_K \mid y_{1:K})$ by initialising $p(x_0 \mid y_{1:0}) = p(x_0)$ and alternating

$$\text{Prediction:} \qquad p(x_{k-1} \mid y_{1:k-1}) \longmapsto p(x_k \mid y_{1:k-1}) \qquad k = 1, ..., K \qquad (26a)$$
$$\text{Update:} \qquad p(x_k \mid y_{1:k-1}) \longmapsto p(x_k \mid y_{1:k}) \qquad k = 1, ..., K. \qquad (26b)$$

Since all operations are linear and all distributions are Gaussian, the recursions are available in closed form. Their parametrisation depends on the parametrisation of Gaussian variables as follows.

### A.1 Covariance-based parametrisation

Let $k = 1, ..., K$. The prediction step computes the predicted distribution $p(x_k \mid y_{1:k-1}) = \mathcal{N}(m_{k|k-1}, C_{k|k-1})$ from the previous filtering distribution $p(x_{k-1} \mid y_{1:k-1}) = \mathcal{N}(m_{k-1|k-1}, C_{k-1|k-1})$ by

$$m_{k|k-1} = A_k m_{k-1|k-1}, \quad C_{k|k-1} = A_k C_{k-1|k-1} (A_k)^\top + B_k. \tag{27}$$

The update step takes the predicted distribution, forms the joint distribution

$$p(x_k, y_k \mid y_{1:k-1}) = p(y_k \mid x_k) p(x_k \mid y_{1:k-1}) = \mathcal{N}(H_k x_k, R_k) \mathcal{N}(m_{k|k-1}, C_{k|k-1}) \tag{28}$$

and computes the update as $p(x_k \mid y_{1:k}) = \mathcal{N}(m_{k|k}, C_{k|k})$,

$$s_{k|k-1} = H_k m_{k|k-1}, \tag{29a}$$

$$S_{k|k-1} = H_k C_{k|k-1} (H_k)^\top + R_k, \tag{29b}$$

$$Z_k = C_{k|k-1} (H_k)^\top (S_{k|k-1})^{-1}, \tag{29c}$$

$$m_{k|k} = m_{k|k-1} + Z_k (y_k - s_{k|k-1}), \tag{29d}$$

$$C_{k|k} = C_{k|k-1} - Z_k S_{k|k-1} (Z_k)^\top. \tag{29e}$$

Iterating these two steps from $k = 1, ..., K$ yields $p(x_K \mid y_{1:K})$.

### A.2 Cholesky-based parametrisation

The Cholesky-based parametrisation predicts the mean like the covariance-based parametrisation. The covariance prediction is replaced by the QR decomposition

$$\mathfrak{Q}\mathfrak{R} = \begin{pmatrix} (L_{C_{k-1|k-1}})^\top (A_k)^\top \\ (L_{B_k})^\top \end{pmatrix} \tag{30}$$

followed by setting $L_{C_{k|k-1}} = \mathfrak{R}^\top$ (Krämer and Hennig, 2024; Grewal and Andrews, 2014). The logic mirrors that in Algorithm 3. The update step in Cholesky-based arithmetic amounts to a QR-decomposition of (Gibson and Ninness, 2005)

$$\mathfrak{Q} \begin{pmatrix} \mathfrak{R}_1 & \mathfrak{R}_2 \\ 0 & \mathfrak{R}_3 \end{pmatrix} = \begin{pmatrix} (L_{R_k})^\top & 0 \\ (L_{C_{k|k-1}})^\top (H_k)^\top & (L_{C_{k|k-1}})^\top \end{pmatrix} \tag{31}$$

followed by setting

$$L_{S_{k|k-1}} = (\mathfrak{R}_1)^\top, \quad Z_k = ((\mathfrak{R}_1)^{-1} \mathfrak{R}_2)^\top, \quad L_{C_{k|k}} = (\mathfrak{R}_3)^\top. \tag{32}$$

Then, use $Z_k$ to evaluate $m_{k|k}$ and iterate $k \mapsto k + 1$. If $p(y_k \mid y_{1:k-1})$ is needed, evalaute $s_{k|k-1}$ like in the covariance-based implementation. Choosing the QR decomposition by Gibson and Ninness (2005) avoids implementing Gaussian updates via Cholesky downdates (e.g. Yaghoobi et al., 2022), which are sometimes numerically unstable (Seeger, 2004). We refer to Krämer (2024, Chapter 4) for why the above assignments yield the correct distributions.

## B Rauch–Tung–Striebel smoother recursions

The Rauch–Tung–Striebel smoother proceeds similarly to the Kalman filter.

### B.1 Covariance-based parametrisation

The prediction step in the smoother involves computing $p(x_k \mid y_{1:k-1})$ like in the filter. It further evaluates

$$p(x_{k-1} \mid x_k, y_{1:k-1}) = \mathcal{N}(G_{k-1|k} x_k + p_{k-1|k}, P_{k-1|k}), \tag{33a}$$

$$G_{k-1|k} = C_{k-1|k-1} (A_k)^\top (C_{k|k-1})^{-1} \tag{33b}$$

$$p_{k-1|k} = m_{k-1|k-1} - G_{k-1|k} m_{k|k-1} \tag{33c}$$

$$P_{k-1|k} = P_{k-1|k-1} - G_{k-1|k} C_{k+1|k} (G_{k-1|k})^\top. \tag{33d}$$

This conditional distribution is stored after every step. The rest of the step proceeds like in the Kalman filter.

### B.2 Cholesky-based parametrisation

The Cholesky-based parametrisation computes the same conditional distribution as the covariance-based parametrisation but replaces matrix multiplication and matrix addition with another QR decomposition; specifically, the QR decomposition (Gibson and Ninness, 2005)

$$\mathfrak{Q} \begin{pmatrix} \mathfrak{R}_1 & \mathfrak{R}_2 \\ 0 & \mathfrak{R}_3 \end{pmatrix} = \begin{pmatrix} (L_{B_k})^\top & 0 \\ (L_{C_{k-1|k-1}})^\top (A_k)^\top & (L_{C_{k-1|k-1}})^\top \end{pmatrix}. \tag{34}$$

This QR decomposition is followed by setting

$$L_{C_{k|k-1}} = (\mathfrak{R}_1)^\top, \quad G_{k-1|k} = ((\mathfrak{R}_1)^{-1}\mathfrak{R}_2)^\top, \quad L_{P_{k-1|k}} = (\mathfrak{R}_3)^\top. \tag{35}$$

This step computes the Cholesky factors of $C_{k|k-1}$ and $P_{k-1|k}$, as well as the smoothing gain $G_{k-1|k}$ in a single sweep. It replaces the prediction step of the Cholesky-based Kalman filter. Compute $m_{k|k-1}$ and $p_{k-1|k}$ like in the covariance-based parametrisation. Store these quantities and proceed with the update step of the Cholesky-based Kalman filter. Again, refer to Krämer (2024, Chapter 4) for why these assignments yield the correct distributions.

## C   Proof of Proposition 1

Recall the notation from Algorithm 2, most importantly, $G_{0|0} = I$, $p_{0|0} = 0$, and $P_{0|0} = 0$. Algorithm 1 computes the transition

$$p(x_0 \mid x_k, y_{1:k-1}) = \mathcal{N}(G_{0|k}x_k + p_{0|k}, P_{0|k}). \tag{36}$$

A forward pass with a Kalman filter gives

$$p(x_k \mid y_{1:k}) = \mathcal{N}(m_{k|k}, C_{k|k}). \tag{37}$$

Then, due to the rules of manipulating Gaussian distributions,

$$p(x_0 \mid y_{1:k}) = \mathcal{N}(G_{0|k}m_{k|k} + p_{0|k}, G_{0|k}C_{k|k}(G_{0|k})^\top + P_{0|k}) \tag{38}$$

follows. To show that this matches the result of Equation 7, it suffices to show

$$p_{0|k} = m_{0|k-1} - G_{0|k}m_{k|k-1} \tag{39a}$$
$$P_{0|k} = C_{0|k-1} - G_{0|k}C_{k|k-1}(G_{0|k})^\top \tag{39b}$$

because the remaining terms already coincide. We use induction to show Equation 39.

**Initialisation**  For $k = 1$, we get

$$p_{0|1} = G_{0|0}p_{0|1} + p_{0|0} = p_{0|1} = m_{0|0} - G_{0|1}m_{1|0}, \tag{40}$$

which shows Equation 39a (compare the smoothing recursion in Appendix B for the last step), as well as

$$P_{0|1} = G_{0|0}P_{0|1}G_{0|0} + P_{0|0} = P_{0|1} \tag{41}$$

which gives Equation 39b.

**Step**  Now, let Equation 39 hold for some $k$. Then, (abbreviate "RTSS": "Rauch–Tung–Striebel smoother")

$$\begin{aligned}
p_{0|k+1} &= G_{0|k}p_{k|k+1} + p_{0|k} & \text{(Algorithm 2)} \\
&= G_{0|k}p_{k|k+1} + m_{0|k-1} - G_{0|k}m_{k|k-1} & \text{(Equation 39a holds for } k) \\
&= G_{0|k}(m_{k|k} - G_{k|k+1}m_{k+1|k}) + m_{0|k-1} - G_{0|k}m_{k|k-1} & \text{(Expand } p_{k|k+1} \text{ as in RTSS)} \\
&= G_{0|k}m_{k|k} - G_{0|k}G_{k|k+1}m_{k+1|k} + m_{0|k-1} - G_{0|k}m_{k|k-1} & \text{(Resolve parentheses)} \\
&= m_{0|k-1} - G_{0|k}(m_{k|k-1} - m_{k|k}) - G_{0|k+1}m_{k|k+1} & (G_{0|k}G_{k|k+1} = G_{0|k+1}) \\
&= m_{0|k} - G_{0|k+1}m_{k+1|k} & \text{(Equation 7)}
\end{aligned}$$

Equation 39a is complete. Similarly,

$$
\begin{aligned}
P_{0|k+1} &= G_{0|k}P_{k|k+1}(G_{0|k})^\top + P_{0|k} &&\text{(Algorithm 2)} \\
&= G_{0|k}P_{k|k+1}(G_{0|k})^\top + C_{0|k-1} - G_{0|k}C_{k|k-1}(G_{0|k})^\top &&\text{(Equation 39b holds for } k) \\
&= G_{0|k}(C_{k|k} - G_{k|k+1}C_{k+1|k}(G_{k|k+1})^\top)(G_{0|k})^\top + C_{0|k-1} - G_{0|k}C_{k|k-1}(G_{0|k})^\top \\
&&&\text{(Expand } P_{k|k+1} \text{ as in RTSS)} \\
&= G_{0|k}C_{k|k}(G_{0|k})^\top - G_{0|k+1}C_{k+1|k}(G_{0|k+1})^\top + C_{0|k-1} - G_{0|k}C_{k|k-1}(G_{0|k})^\top \\
&&&\text{(Expand parentheses, } G_{0|k}G_{k|k+1} = G_{0|k+1}) \\
&= C_{0|k-1} - G_{0|k}(C_{k|k-1} - C_{k|k})(G_{0|k})^\top - G_{0|k+1}C_{k+1|k}(G_{0|k+1})^\top &&\text{(Rearrange)} \\
&= C_{0|k} - G_{0|k+1}C_{k+1|k}(G_{0|k+1})^\top &&\text{(Equation 7)}
\end{aligned}
$$

the covariance recursion must hold. Equation 39 holds for all $k \geq 1$, and the statement is complete.

