# OpenReview forum: "Numerically Robust Fixed-Point Smoothing Without State Augmentation"
_TMLR — Accepted by TMLR_

### Review · Reviewer_UyAV · 2024-09-06

**Summary Of Contributions:**

The authors, by joggling the order of marginalisation and factorisation in filtering equations, devise an algorithm which appears to shift the pareto frontier of efficiency and stability for the problem of identifying starting conditions in a filter problem

**Audience:**

Yes

**Claims And Evidence:**

Yes

**Requested Changes:**

Minor clarity issue: sec 2.2, the intro paragraph refers to two things which don't match exactly the following two headings:

>The state of the art in fixed-point smoothing does not use fixed-interval smoothers to compute p(x0 | x1:K ).
>There are two more efficient approaches: Either we solve the fixed-point smoothing problem by computing
>the solution to a specific high-dimensional filtering problem (Biswas and Mahalanabis, 1972), or we derive
>recursions for how p(x0 | y1:k) evolves with k = 1, ..., K (Meditch, 1967b;a; 1969).

Would that be better rewritten as the following?

>The state of the art in fixed-point smoothing does not use fixed-interval smoothers to compute p(x0 | x1:K ).
>There are two more efficient approaches: Either we solve the fixed-point smoothing problem by computing
>the solution to a specific high-dimensional filtering problem (_state-augmented filtering_, Biswas and Mahalanabis, 1972), or we derive
>_fixed-point recursions_ for how p(x0 | y1:k) evolves with k = 1, ..., K (Meditch, 1967b;a; 1969).

Or have I misunderstood?

There is another weird problem with flow in section 2.2.

>**Fixed-point recursions** Suppose we temporarily ignore Cholesky-based forms and commit to covariance-
>based parametrisations of Gaussian distributions. In that case, we can improve the efficiency of the fixed-point
>smoother

I think you can delete the bits about CHolesky because in this point in the paper, while w have described THAT the cholesky decompositon can be used, we haven't actually seen it in action; everything so far has been in terms of covariances.

>**Fixed-point recursions** We can improve the efficiency of the fixed-point
>smoother in terms of covariances

7a: can you clarify? set k=1. Then we have

$$
\begin{aligned}
G_{0|1}&=G_{0|0}G_{0|1}\\\\
&=G_{0|1}\\\\
\end{aligned}
$$

This looks under-specified; Anything times the identity is itself. What is going wrong here? Have I introduced an off-by-one error? Should $G$ be coupled with $C$? Something else?

**Strengths And Weaknesses:**

# Strengths

* simple method that solves a well-defined problem neatly
* beautiful explanation of the filter recursions, including square root form, which is surprisingly rare in the literature
* simple but challenging example problems

# weaknesses

* occasionally unclear text (see requested changes)
* the authors miss one easy "win": as they point out, their innovation allows arbitrary Gaussian parameterisation of the fixed point recursion. If their goal to to minmise cost *and* stability, they should probably characterise the efficiency of the information parameterisation also, since that parameterisation can lead to efficient results some times.

---

> ### Author Response · Authors · 2024-10-24
>
> Thank you so much for your positive assessment. We're glad you enjoyed the paper!
> Below, we will reply to the specific points you raised.
> We're looking forward to hearing what you think about the changes.
>
>
>
>
> ## Strengths and weaknesses
> Thank you so much for listing all those strengths. We hope that you continue fighting for this paper's acceptance!
>
> As for the "missed easy win," yes, we agree that the information form would be an interesting extension. We did not investigate information forms in this paper because, at this point, we don't have a concrete need for them. However, since z28E asks a similar question, we'll update the conclusion: We look forward to future work discussing the information form and other parametrisations of Gaussian variables in the context of numerically robust fixed-point smoothing. Thank you for that comment.
>
>
>
> ## Requested changes
>
> > Minor clarity issue: sec 2.2, the intro...
>
> Are you referring to directly linking those methods to the paragraph headings that follow? That is a good idea, we'll change the sentence accordingly.
>
> > I think you can delete the bits about CHolesky because...
>
> Thanks for that point. Would your final recommendation depend on this change? We'd prefer to keep the current version because, unlike the state-augmentation approach, which has covariance- and Cholesky-based forms, the fixed-point recursions have previously only been known for covariance-based arithmetic. Cholesky-based codes are our contribution. However, if you have a suggestion of how to improve the phrasing to clarify this point, we'd be happy to reword it!
>
>
> > This looks under-specified; Anything times the identity is itself
>
> This behaviour is on purpose; the first fixed-point backward transition matches the first smoothing backward transition by definition. The parametrisations only differ in later steps. Another way of expressing Equation 7 would be as "if k=1, do nothing and store the smoothing parameters as the fixed-point solution; if k>1, implement Equation 7", but we found our formulation of initialising with the identity matrix and then always using Equation 7 to be more concise. It also matches how one would implement Equation 7 in software. We've added a sentence to the presentation of Equation 7 to reflect this initialisation. Does that help?
>
> Again, thank you so much for reviewing this paper so positively. We look forward to the discussion and hope you continue arguing for accepting this work.

---

> ### Comment · Reviewer_UyAV · 2024-10-24
>
> Sec 2.2:
>
> >  I think you can delete the bits about CHolesky because...
>
> Ah yes. I could have said that better: my point is not deleting the reference to choleksy decomp entirely, but about the flow of the paper; up to this point we've _mentioned_ the square root form but only shown classic-style covariance operations in the Kalman filter. It makes sense, but feels is a little clunky to me, and not quite self-contained, because the reader has to know what is going on (I suspect reviewer z28E's comments may be bit about this also). We get explicit introductions to the square root form only later in the paper (sec 3.2) where you introduce it for your method in particular. As I said, I think _that_ explanation is unusually well done, it's just the order of introducing the concepts.
>
> tl;dr I'm not a huge fan of this order stylistically, but it is not an actual blocker, just my feedback on the pedagogic flow.
>
> >  This looks under-specified; Anything times the identity is itself
>
> yesssss OK I get that. I agree this is defensible notation, definitely include the explanation though for n00bs like me seeing it for the first time

---

> > ### Author Response · Authors · 2024-10-25
> >
> > Thanks for iterating!
> >
> > We understand your perspective about the order of topics; thanks for the feedback! However, since it doesn't seem to be a blocking issue, and since it would be a relatively extensive change at this point, we'd prefer to keep the order of paragraphs as is. But perhaps we can improve the presentation regardless: the "Kalman filtering" paragraph in Section 2.2 discusses covariance- and Cholesky-based implementations and links to Appendix A for details. We could emphasize this point by replacing
> >
> > > Appendix A contrasts Cholesky-based and covariance-based Kalman filtering
> >
> > with
> >
> > > Appendix A contrasts how Cholesky-based and covariance-based Kalman filtering implementations differ concretely. Later sections extend this contrast to fixed-point smoothing; Cholesky-based fixed-point smoothing is the main contribution of this work.
> >
> >
> > Do you think a replacement like the above would improve the flow?

---

### Review · Reviewer_TN7Q · 2024-10-20

**Summary Of Contributions:**

This work introduces a new approach for fixed point smoothing that generalizes existing techniques such as the Racuh-Tung-Striebel smoother, filtering via an augmented state, and fixed point recursions. This is achieved by producing a new recursive derivation for fixed point smoothing, and carefully deriving update rules that can avoid an eugemnted state space computation. This work concludes with a variety of experiments designed to show the computational advantages of this new methodology.

**Audience:**

Yes

**Broader Impact Concerns:**

No broader impact concerns.

**Claims And Evidence:**

Yes

**Requested Changes:**

Following up the above comments I would like to see the introduction improved and contain a richer set of related works and background for the reader.

**Strengths And Weaknesses:**

This work is well written and tackels an important problem. The contributions listed in the manuscript seem at first glance correct and meaningful. I am by no means an expert in this literature, so I may not be the best person to assess their relevance within it. This is my main concern, that the derivations in this work, although correct, may be either existing in the literature, or very derivative. That being said, the paper makes a good point in convincing the reader their results are correct. One of my main concerns is the presentation of related and historical work. The introduction is extremely short and devles into technical details almost immediately, this is an issue in my opinion because it does not give the reader any bird's eye view of the field.

---

> ### Author Response · Authors · 2024-10-24
>
> Thank you for taking the time to review and verify our approach's correctness!
> Thank you further for acknowledging our manuscript is well-written and tackles an important problem.
> In the following, we'll reply to your requested changes and what you point out as weaknesses.
> We look forward to the discussion!
>
>
>
> ## Strengths and weaknesses
> > "This is my main concern, that the derivations in this work, although correct, may be either existing in the literature, or very derivative."
>
> Yes, we agree with the assessment that our paper solves a specific problem (Cholesky-based implementation) within a particular field (state-space models, filtering, smoothing, fixed-point smoothing); concretely, we directly extend the method by Särkkä and Hartikainen.
> However, to the best of our knowledge, our results have been previously unknown. Please let us know if you believe the current version is missing something. We'd be happy to update the manuscript!
>
>
> > "One of my main concerns is the presentation of related and historical work. The introduction is extremely short and devles into technical details almost immediately, this is an issue in my opinion because it does not give the reader any bird's eye view of the field."
>
> Thank you for raising this.
> The motivation for a brief introduction and then discussing all related work as a part of and after the problem statement was that distinguishing related papers requires some context: We found it easiest to distinguish filtering from fixed-interval and fixed-point smoothing after introducing the state-space model because the differences can be made more precise.
> However, we hear your criticism! Would you like us to add some information to the introduction about the role of state-space models, filtering, and smoothing algorithms in contemporary machine learning? We agree with you that this might lower the entry barrier for some readers. What do you think? Would this help?
>
>
> ## Requested changes
>
> > Following up the above comments I would like to see the introduction improved and contain a richer set of related works and background for the reader.
>
> We assume our reply above covers these points. If not, please let us know, and we will revisit this question.
>
>
>
> Again, thanks so much for reviewing this paper so positively! We would be happy to hear your thoughts on our replies and look forward to the discussion.

---

### Review · Reviewer_z28E · 2024-10-20

**Summary Of Contributions:**

This work addresses the fixed-point smoothing problem, where the goal is to determine the posterior distribution of the initial state given all observations, specifically $p(x_0 \mid y_{1:K})$. This problem is particularly important for dynamical systems with unknown initial conditions. The authors focus on a linear, discrete-time state-space model with additive Gaussian noise and present a numerically robust solution based on Cholesky factorization.

A key advantage of this approach over the previous method, "fixed-point smoothing via space-augmented filtering," is that while both rely on Cholesky factorization, the new method eliminates the need for state augmentation, which reduces both runtime and memory usage.


This work is closely related to the "fixed-point recursions" method by Särkkä and Hartikainen (2010), which introduced covariance-based recursions for the fixed-point smoothing problem. A key benefit of Särkkä and Hartikainen's method is that it avoids doubling the size of the state-space model, resulting in lower memory usage and faster runtime. Building on this, the current work proposes a Cholesky-based formulation of the "fixed-point recursions" method, maintaining the same efficiency in terms of memory usage and runtime.
All derivations for the proposed method (Algorithms 1 through 3) have been verified and found to be correct.

The numerical examples implemented using JAX effectively demonstrate and support the theoretical claims.

Reference:

Simo Särkkä and Jouni Hartikainen. On Gaussian optimal smoothing of non-linear state space models. IEEE
Transactions of Automatic Control, 55(8):1938–1941, 2010.

**Audience:**

Yes

**Broader Impact Concerns:**

There are no substantial concerns that necessitate a broader impact discussion.

**Claims And Evidence:**

Yes

**Requested Changes:**

Major issues:
1. After Equation 7, it is noted that these formulas do not have a Cholesky-based formulation. Does this imply that it is not possible to derive Cholesky-based formulations directly from these equations, rather than relying on Algorithms 1 and 3?

2. Since this work focuses on linear discrete-time state-space models and claims that it can be further generalized to nonlinear models through techniques such as posterior linearization, it is essential to derive closed-form Cholesky-based formulations for all steps involved in finding $p(x_0 \mid y_{1
})$ using Algorithms 1 and 3, based on the state-space model in Equation 1 and the Cholesky form of the covariances in Equation 2. For instance, what is the formulation for $L_{P_{k-1 \mid k}}$ in terms of the model parameters?

Minor issues:
1. In Equation 2, $b_k \sim \mathcal{N}(0, B_k)$ is given. What is the difference between B_k and Q_k which introduced above Equation 1?
2. Above Equation 3; Kalman filter computes $p(x_k \mid y_{1:k})$ for $k = 1, \ldots, K$ not only $p(x_K \mid y_{1:K})$.
3. Feel free to use the correct name "Särkkä" instead of "Sarkka" for the reference Simo Särkkä and Jouni Hartikainen (2010).
4. Regarding Equation 11, it is worth noting that $p(x_0 \mid x_K, y_{1:K}) = p(x_0 \mid x_K, y_{1:K -1} ) $, that is, given $x_K$, $x_0$ is independent from $y_k$, which we also see through rearranging the integrals.
5. Step 2 of Algorithm 1: the iteration is from $k = 1$ to $k = K$. What does iterate from $k-1$ to $k$ mean, it seems to be extra.
6. In Step 3 of Algorithm 1, there is a missing parenthesis in  $p(x_0 \mid y_{1:K})$.
7. Section 3.2, in the second line, $p(x_0 \mid x_k , y_{1:k})$ should be $p(x_0 \mid x_k , y_{1:k-1})$.
8. In Algorithm 2 and 3, the notation $N$ is used instead of $K$.
9. The subscript in Equations (16c) and (16d) should be $0|k$.
10. If I understand correctly, all the methods in Tables 3 and 4 are Cholesky-based. If so, please indicate this in the tables.
11. In the first experiment, what is the justification for using only three runs? Is that sufficient? Additionally, why not use the average runtime instead of selecting the best one?

**Strengths And Weaknesses:**

In the previous section, I outlined the advantages of this work over prior approaches. However, I believe the importance of addressing the fixed-point smoothing problem has not been sufficiently emphasized. Additionally, since this work is closely related to the fixed-point recursions method, particularly equations (7a) to (7c), further explanation is needed as to why the authors did not derive the Cholesky formulations directly from these equations. More explanations will be given in the next section.

---

> ### Author Response · Authors · 2024-10-24
>
> Thank you for your positive assessment and for believing in this submission's strengths. Your concrete feedback has been very helpful!
> Below, we reply to the requested changes and to what your review points out as weaknesses.
>
> ## Strengths and weaknesses
>
> > importance of addressing the fixed-point smoothing problem
>
> Thanks for pointing this out.
> The introduction mentions Meditch's work, and the experiments demonstrate applications to probabilistic numerics and expectation maximisation.
> We consider probabilistic numerics especially relevant to fixed-point smoothing in the future because, for example, ODE inverse problems commonly involve unknown initial conditions.
> We agree that this discussion would improve the motivation of fixed-point smoothing and will add it to the paper.
>
>
> > equations (7a) to (7c), further explanation is needed as to why the authors did not derive the Cholesky formulations directly from these equations
>
> That's a good question. See our reply to "Requested changes" below.
>
>
> ## Requested changes
>
> > After Equation 7, it is noted that these formulas do not have a Cholesky-based formulation.
>
> Thanks for pointing out that "it does not have a Cholesky-based formulation" could be misleading. The sentence in the paper means that a Cholesky-based version of these formulas has been previously unknown. We'll rephrase the sentence to reflect that connotation. We hope you agree with us that this change clarifies the presentation!
>
>
> > Does this imply that it is not possible to derive Cholesky-based formulations directly from these equations
>
> When answering this question, we assume that by "directly," you mean a linear algebra-based approach (which Särkkä and Hartikainen take; see Section 10.4 in Simo Särkkä's "Bayesian filtering and smoothing".) as opposed to our probability-density-based approach. If your question targets something else, please clarify, and we will revise our answer!
>
> Proposition 1 states and proves that the combination of Algorithms 1 and 2 recovers the recursion in Equation 7. From this perspective, our method directly generalises Särkkä and Hartikainen's fixed-point recursion. We prefer our approach of manipulating probability distributions over manipulating matrix recursions because there is a clear path for going beyond Cholesky-based formulas: our method also opens the door for ensemble- or information-based parametrisations by implementing alternatives to Algorithms 2 and 3. The conclusion in Section 5 and the end of the "Notation" paragraph in Section 1 discuss this. Still, since AyUV asked for a similar clarification, we'll add a sentence to the conclusion discussing other Gaussian parametrisations. Does this answer your question?
>
>
> > and claims that it can be further generalized to nonlinear models through techniques such as posterior linearization
>
> The paper does not claim that our method can be generalised to nonlinear models. Instead, we point this out as potential future work. We hope we didn't make this claim; if the article contains an unfortunate phrase that may suggest this (we double-checked and couldn't find any), please let us know, and we will correct it!
>
>
> > it is essential to derive closed-form Cholesky-based formulations for all steps involved in finding using Algorithms 1 and 3, based on the state-space model in Equation 1 and the Cholesky form of the covariances in Equation 2. For instance, what is the formulation for in terms of the model parameters?
>
> When answering this question, we assume that you're asking about where the parameters in Equation 15 come from: 15a is from the previous step, and 15b comes from the square-root implementation of the Rauch--Tung--Striebel smoother (Equation 4; a precise formula for L is in Appendix B). We'll update the presentation of Equation 15. Thanks again for this comment.
>
>
> About the minor issues, thanks for catching the typos! We only reply to the non-typo points below:
>
> 4. Yes, this is correct. However, could you maybe clarify whether you would like us to make a specific change?
> 5. The "iterate" means that we must store the $k-1$th step to parametrise the $k$th step. It indicates how this loop must be executed sequentially (not in parallel).
> 11. Good catch! The experiments accidentally omitted this information (we'll add it). The differences between runs were minimal, so we found three sufficient. We report the fastest of three runs instead of the average (both are popular) for the following reason: The algorithm is entirely deterministic (no data adaptivity and no randomness), so all three runs implement precisely the same operations. Thus, the main driver for runtime differences is background machine noise. Choosing the best of three runs removes it as much as possible.
>
>
> Again, thank you for reviewing this paper so positively and for your concrete suggestions. We hope we were able to address your concerns adequately. We look forward to the discussion!

---

### Author Response · Authors · 2024-10-24

Thank you to all reviewers for their time and for the positive assessment!
It seems that most of the mentioned weaknesses are specific questions about specific sentences, and we have answered all of them in detail below.

We've now uploaded a revised PDF, which includes the following changes:
- Fix the typos pointed out by z28E.
- Emphasise probabilistic numerics as an application for fixed-point smoothing, as z28E asked.
- Rephrase the sentence "it does not have a Cholesky-based formulation", as pointed out by z28E.
- Explain where the parameters in Equation 15 come from, as inquired by z28E.
- Justify the choice of "fastest of three runs" as a runtime metric, as requested by z28E.
- Discuss information form and other parametrisations in the conclusion as pointed out by z28E and UyAV.
- Link methods to paragraph headings in Section 2.2 as suggested by UyAV.
- Clarify the initialisation of Equation 7, as UyAV asked.
- Discuss state-space models' roles in contemporary machine learning, as requested by TN7Q.

We hope these changes resolve your concerns. In any case, we look forward to further discussion!
Again, thank you all for reviewing this paper.

---

### Decision · Action_Editor_3Byh · 2025-01-14

**Recommendation:** Accept as is

**Comment:**

The reviewers all agreed that the claims were well-supported and there was a sufficient audience for the work. During the discussion period, the paper was revised by the authors both for clarity and to motivate the need for this work. As there are no major outstanding concerns from the reviewers, I recommend acceptance as is.

**Audience:**

The fixed-point smoothing problem is arguably somewhat narrow. However, as pointed out by the authors and confirmed by reviewers, the field of probabilistic numerics and in particular those interested in e.g. ODE inverse problems will likely find the results of this paper timely and useful. Therefore, the criterion for an audience at TMLR is satisfied.

**Claims And Evidence:**

This paper proposes a novel algorithm for the fixed-point smoothing problem, the goal of which is to produce a posterior distribution for the initial state of a state-space model. This stands in contrast to fixed-interval smoothing, which aims to produce a posterior for the entire sequence of states. The proposed algorithm is claimed to have superior computational efficiency and numerical robustness compared to previous approaches. The theoretical properties of the proposed algorithm are analyzed in Section 3. The efficiency and robustness benefits are empirically confirmed in Section 4. Overall, the claims are clearly stated and are supported with theoretical and experimental evidence.